# Air Quality and the Spatial-Temporal Differentiation of Mechanisms Underlying Chinese Urban Human Settlements

**Xueming Li [1,2], Songbo Li [1,2,\*], Shenzhen Tian [1,2], Yingying Guan [1,2] and He Liu [1,2]**

1   School of Geography, Liaoning Normal University, 850 Huanghe Rd., Dalian 116029, China;
    lixueming@lnnu.edu.cn (X.L.); tsz999@lnnu.edu.cn (S.T.); gyy9418@126.com (Y.G.); liuhe1581@163.com (H.L.)
2   Human Settlements Research Center, Liaoning Normal University, 850 Huanghe Rd., Dalian 116029, China
*   Correspondence: 0825lisongbo@sina.com; Tel.: +86-188-4080-1683

**Abstract:** Urban air has its typical structural characteristics. With the continuous optimization of urban human settlement indicators, the key issue and single system of "air quality" in urban human settlements needs to be further discussed. Based on air conditions, this paper attempts to visually measure the spatial-temporal distribution of human settlements in 283 prefecture-level cities in China using ArcGIS and Matlab and tries to reveal the influencing mechanisms: (1) There is no significant difference between the average of the comprehensive score of human settlements in 6 years. The overall level of those in all cities decreases from 0.6581 to 0.6004 year by year, and the average level order in the seven regions of China is Southern China (0.7310) > Southwest China (0.6608) > East China (0.6515) > Northeast China (0.6496) > Northwest China (0.6049)> Central China (0.5901) > North China (0.5565). (2) The global *Moran's I* index of China's human settlements is between 0.3750–0.7345, showing a positive spatial correlation, and the comprehensive development level has the characteristics of local spatial convergence of low-value clusters in the middle and lower reaches of the Yellow River and high-value clusters in the south coast and Heilongjiang Province. (3) The spatial econometric model tests the influencing mechanism. There is a significant spatial positive correlation between science and technology investment in each city. The urbanization rate, the degree of advanced industrial structure, and the urban average elevation have a certain spatial spillover, showing a negative correlation. Science and technology investment and the degree of advanced industrial structure have the greatest impact.

**Keywords:** air quality; human settlements; spatial-temporal differentiation; spatial econometric models; China's prefecture-level city

## 1. Introduction

According to Maslow's hierarchy of needs, the first level need emphasizes the basic survival and life of human beings, including resources of air, water, ecology, and environment. Compared with water and food, air is the biggest consumable of the human body. Air is needed by everyone; its quality not only affects people's health, but is also the most intuitive and basic factor for them in choosing concentrated settlements [1,2]. Human settlements are a complex system, and human dwelling districts are places where complex systems interact with human beings [3]. UN-Habitat reports that by 2050, two-thirds of the world's population is expected to live in urban areas. Cities can efficiently accommodate a large number of people in a relatively limited space, and the quality of urban air can directly affect residents' health, economic investment, and social development [4,5]. Therefore, it is necessary to adjust the urban base intuitively and directionally through the research on human settlements based on air quality, so as to make new progress in ecological civilization as well as achieve sustainable, green, and sound urban development during the 14th Five-Year Plan.

Studies on how to improve the quality of urban residents' lives, and the influencing factors of residential areas, have become the main focus of livable city research since 1970.

After 1980, with the continuous improvement of the idea of sustainable development, Georges and others have studied indicator systems of urban sustainable development in Western developed countries, and proposed the selection strategy of sustainable development indicators [6]. Henry Rasch believes that we should focus on the long-term livability of cities, and the nature of a livable city proposed by Kaserati includes "living" and "ecological" sustainability [7]. Timothy analyzes it from the perspective of ordinary citizens and believes that a better environment for residents is one of the factors, and Evans proposes that "livability" includes two definitions: one is suitable for residents, and the other is meeting the requirements of sustainable development [8]. In the *Introduction to Human Settlement Science*, Wu Liangyong proposes that a "sustainable and pleasant living environment is the goal of human settlements science". In the research of geography and human settlements, we must take sustainable development as the link [3].

Dahiya links the new urban agenda with key global sustainable and inclusive urban development issues, forecasting emerging trends in urban sustainability [9]. Shaparev uses annual data on environment in sustainable development indicators developed in the United States and Russia [10]. Corbane uses data, such as the global human settlements urban center database, to quantify and analyze the changes in greenness in urban centers, which are very important to achieve the UN sustainable development goal [11]. Scholars studying sustainable development from different perspectives of human settlements, embody the concept of creating a low-carbon or even zero-carbon emission city in practice, and have finally turned to a new field: creating a sustainable city. In order to optimize the regulation and control of the sustainable development of urban human settlements, it is necessary to establish urban human settlements evaluation indicator system. Because it is an important basis for the comprehensive evaluation of the sustainable development stage, degree, and quality of urban human settlements [5].

Scholars at home and abroad use different methods to explore the distribution of air quality [12,13] and analyze the dynamic trends and change rules of different air pollutants [14]. Natural environmental factors, such as topography, green space coverage, and meteorological conditions, can have a certain impact on air quality in urban areas [15]. Direct consumption of fossil fuels and transportation energy [16,17] and urban industrial scale and structure [18] can increase the concentration of pollutants. In addition, urban population size [19], age characteristics [20], family types [21], and urbanization levels [22] also have different effects on pollution emissions and restrict the development pattern of settlements [23]. The above air research exists in many residential environment evaluation systems: Wu calculates the regional climate comfort index by using meteorological conditions, such as air temperature and relative humidity [24]. Yang establishes the comprehensive evaluation system of green human settlements and ecological environment elements, including atmospheric, urban form, and temperature environment [25,26]. Tang constructs a comprehensive evaluation system focusing on living conditions, urban ecological environment, and social economy [27]. Sykes attaches importance to all aspects of urban livable environment, including the environment of socio-economy, living, and ecology [28].

The balance between the built residential environment and the natural environment is a key link in urban planning and construction, and it is also an important foundation and guarantee for improving the well-being of residents [29–31]. Evaluation of urban human settlements at home and abroad mostly uses the comprehensive index system to measure the regional advantages and disadvantages. Among them, the meteorological data of urban human settlements are important for sustainable development [32] because they can quantify and analyze the change of urban human settlements greenness, as well as evaluate the urban disaster resilience and urban environmental cleanliness [11]. Moreover, the atmospheric environment also exists in the evaluation factor of urban systems [33], which can quantitatively evaluate the quality and the livability of urban natural and cultural environments [34,35], theoretically guiding the construction of a livable city [36].

With the progress of geography and related disciplines, many modern remote sensing and computer technologies are applied to human settlements assessments to provide basic support for environmental quality assessment. In recent years, with the combined data of satellite image, space, and other multi-sources as the frontier, mainstream methods, such as fuzzy mathematics, fuzzy measure theory, and vector operator have been adopted. Useful information such as disaster prevention and planning, surrounding regional linkage, and management policies have been determined, promoting the establishment, evaluation, and model discussion of human settlements system indicators [37–39]. The correlation between air quality and surrounding areas makes the link between human settlements and the surrounding cities attract people's attention [40,41]. Scholars at home and abroad combine traditional and emerging methods and technologies, formulate evaluation indicators, and establish mathematical models to conduct research, evaluation, and prediction of spatial-temporal differences [42,43]. They focus on urban planning and future research from the perspective of livability and the sustainability of human settlements, so as to study the relationship between air quality and urban health [43,44].

At present, there are few studies on examining an important issue from a single perspective, formulating targeted policies and measures to serve a wide range of social groups in order to protect existing urban bases [44–46]. To sum up, in order to effectively improve the health of residents and the quality of living atmosphere, as well as solve the problems of human settlements, this paper takes the basic needs of urban residents as the core and selects the air-oriented evaluation system, which is a unique indicator combination type, in order to study the spatial-temporal differentiation of human settlements in 283 prefecture-level cities in China. It also analyzes these influencing factors through the spatial dependence and spillover effect of geographical elements, so that decision-makers can have the most intuitive understanding of the distribution and determinants of air livable cities, and provide a new perspective for the subsequent evaluation of urban livable environments.

## 2. Data Sources and Research Methods

### 2.1. Data Sources

According to the principle of spatial consistency and the comparison of administrative division directories published by the Ministry of Civil Affairs over the years, 283 prefecture-level Chinese cities in 2013 are consolidated as sample areas (Figure 1). In order to explore the spatial-temporal differentiation of human settlements and influencing factors from 2013 to 2018, the data used consider the scientificity, availability, and practicability of the indicators, and data sources are as follows (Table 1). Some missing values are completed by SPSS regression analysis.

### 2.2. Research Methods

#### 2.2.1. Weight Assignment

As an objective weight method, CRITIC determines the objective weight of indicators based on two basic concepts. The first is contrast strength between various indicators. The standard deviation is used to show the gap between values of indicators in multiple-evaluation schemes. There is a positive correlation between the value of standard deviation and the degree of the gap. The second is conflict among indicators. The index weight is calculated through index data fluctuation or their correlation. If there is an obvious positive correlation, it shows that their conflicts are not obvious [47]. AHP combines qualitative analysis with quantitative analysis. It mainly depends on the experience of decision-makers to judge and distinguish relative importance of indicators to give reasonable weights to various indicators. It also sorts the advantages and disadvantages of various schemes according to the size of the weights [39].

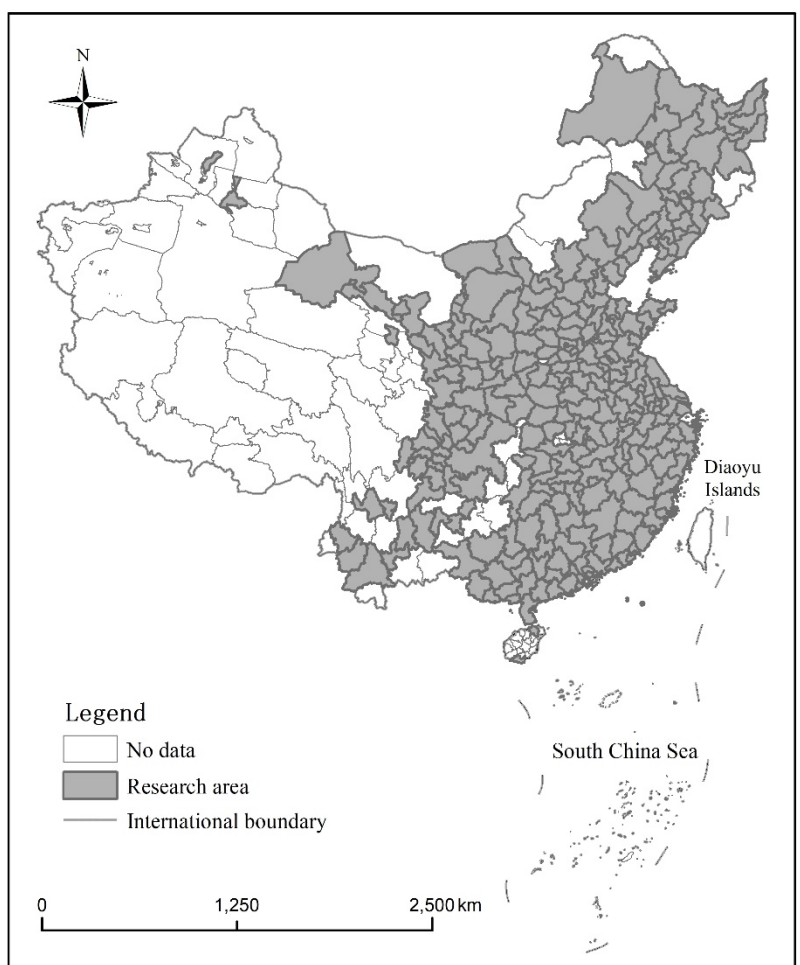

**Figure 1.** Study area.

**Table 1.** Data sources (accessed on 2 September 2021).

| Data Type | Specific Variables | Data Sources |
|---|---|---|
| Administrative map | Urban planning | Standard map service system (http://bzdt.ch.mnr.gov.cn/) |
| Air quality monitoring data | AQI, $SO_2$, $NO_2$, $PM_{2.5}$, $PM_{10}$, CO, $O_3$ | 1. Air quality online monitoring and analysis platform (https://www.aqistudy.cn/); 2. Air quality history data (https://aqicn.org/city/dalian/cn/) 3. Atmospheric Composition Analysis Group: Surface $PM_{2.5}$ (http://fizz.phys.dal.ca/~atmos/martin/?page_id=140) 4. Earth Observing System Data and Information System (https://sedac.ciesin.columbia.edu/data/set/sdei-global-annual-gwr-pm2-5-modis-misr-seawifs-aod/data-download) |
| Meteorological data | Temperature, rainfall, wind speed, and humidity | National Meteorological Science Data Center (https://data.cma.cn/data/cdcindex/cid/0b9164954813c573.html) |
| Urban social and economic development index data | Evaluation index: highway freight volume, green coverage rate, second output value, etc. Driving mechanism: urbanization rate, industrial structure, education expenditure, etc. | Urban Statistical Yearbook, Water Resources Bulletin, Environmental Quality Report, China Environmental Statistics Yearbook, China Energy Statistical Yearbook, Bulletin on National Economic and Social Development (https://data.stats.gov.cn/; https://navi.cnki.net/knavi/yearbooks/index) |

| Data Type | Specific Variables | Data Sources |
|---|---|---|
| Topographic relief | Average urban elevation | Geospatial data cloud (http://www.gscloud.cn/sources/index?pid=302&ptitle=DEM%20%E6%95%B0%E5%AD%97%E9%AB%98%E7%A8%8B%E6%95%B0%E6%8D%AE&rootid=1) DEM digital elevation data, with a resolution of 30 m |
| Residential activity data | Population density and resident activity intensity | 1. Fifth and sixth census data (https://navi.cnki.net/knavi/yearbooks/index) 2. Night light value (https://www.ngdc.noaa.gov/eog/dmsp.html) 3. *Urban Statistical Yearbook* |

This paper makes a comprehensive evaluation with research based on the volatility, correlation, and size of data. Using CRITIC and AHP, the calculation results are fused to obtain the optimized index weight. It can effectively reduce the one-sidedness of using only one method to determine the weight and make up for the lack of information reflected by the two (Table 2).

The analytic weight of AHP is $B_j$, and the objective weight is $W_j$:

$$U_j = \frac{W_j B_j}{\sum_{i=1}^{n} W_j B_j} \tag{1}$$

**Table 2.** Comparison of the CRITIC method and AHP method.

| Comprehensive Evaluation | Data Fluctuation | Correlation between Data | Figure Size |
|---|---|---|---|
| AHP | No | No | Yes |
| CRITIC | Yes | Yes | No |

Human settlements cover the natural and man-made sources of air, which coincide with the natural and humanistic elements. Based on this, this paper considers the elements of residents' perception of air conditions, and determines the natural and humanistic elements as the indicators for evaluating the quality of human settlements. When selecting the factors, the scientificity, availability, and practicability of them are mainly considered. Based on the research of domestic and foreign scholars, two primary indicators, five secondary indicators, and twenty-five tertiary indicators are preliminarily established. The indicators are as follows (Table 3):

**Table 3.** Evaluation indicators and comprehensive weight of human settlements based on air quality from 2013 to 2018.

| Target Layer | Criterion Layer | Index Level | Criterion Attribute |
|---|---|---|---|
| Natural environment | Meteorologic condition | Annual average temperature (°C) | * |
| | | Average annual relative humidity (%) | * |
| | | Average annual rainfall (mm) | + |
| | | Mean wind speed (m/s) | * |
| | Air pollutants | AQI | − |
| | | $PM_{2.5}$ ($\mu g/m^3$) | − |
| | | $PM_{10}$ ($\mu g/m^3$) | − |
| | | $NO_2$ ($\mu g/m^3$) | − |
| | | $SO_2$ ($\mu g/m^3$) | − |
| | | CO ($\mu g/m^3$) | − |
| | | $O_3$ ($\mu g/m^3$) | − |

**Table 3.** *Cont.*

| Target Layer | Criterion Layer | Index Level | Criterion Attribute |
|---|---|---|---|
| Cultural environment | Air control | Green and cover rate in the built-up area (%) | + |
| | | Per capita park green area (m$^2$) | + |
| | | Number of days with good or above Grade 2 air quality (days) | + |
| | | Industrial smoke (powder) dust treatment rate (%) | + |
| | Economic development | Highway passenger Volume (10,000 persons) | − |
| | | Highway freight volume (10,000 tons) | − |
| | | Number of operating vehicles (vehicles) with bus (electric) vehicles | + |
| | | Construction of urban housing (10,000 square meters) | − |
| | | The second output value accounted for the GDP proportion (%) | − |
| | | Urban population density (people/km$^2$) | − |
| | Energy consumption | Total gas supply (artificial and natural gas) (ten thousand cubic meters) | − |
| | | Total LPG gas supply (ton) | − |
| | | Dust industrial dust emission per capita (ton) | − |
| | | Per capita industrial sulfur dioxide emissions (ton) | − |

Note: The nature of each indicator is relative to the evaluation target, + refers "positive", and higher values mean the better. − refers "negative", and lower values mean the better. * refers to moderation, and moderate values are fine. When the index value is less than the moderate value, it conforms to a positive index. Moreover, when it is greater than the moderate value, it conforms to an inverse index.

### 2.2.2. Spatial-Temporal Differentiation Measurement Model

Global spatial autocorrelation is used to analyze the overall spatial distribution mode and state of urban human settlements in China. It can accurately reflect whether there are random, clustered, or discrete spatial distribution between cities and their surrounding areas. This paper uses the *Moran's I* index to measure whether there is autocorrelation of human settlements in prefecture-level cities in China [48].

Global autocorrelation only evaluates the overall state of the investigation object, but it cannot reflect the specific correlation between each region and its surrounding adjacent regions. In order to intuitively reflect the spatial correlation of local research objects, it is necessary to use ArcGIS spatial clustering and outlier methods (*Anselin Local Moran's I*) to analyze China's urban human settlements, so as to intuitively observe the agglomeration state of local regions [49].

### 2.2.3. Calculation Method of Influencing Factors

Based on the basic principles of spatial geography, the basic ideas of geography are applied to the study on regional practical problems, highlighting the spatial effect in the econometric model. Firstly, the spatial autocorrelation of human settlements is tested to determine whether it is necessary to expand the time series data of influencing factors into a spatial econometric model. Common models include Spatial Lag Model (SLM), also known as Spatial Auto-regressive Model (SAR), Spatial Error Model (SEM) and Spatial Durbin Model (SDM) [50].

(1) Spatial Lag Model (SLM): if there is a substantial correlation between geographical elements, such as inter-regional economy, terrain, etc., it can be analyzed by adding the spatial lag factor of the dependent variable. All explanatory variables will directly act on dependent variables through the spatial transmission mechanism.

$$Y_{i,t} = \alpha + \rho \sum_{j=1}^{N} W_{i,j} Y_{i,t} + \beta X_{i,t} + C_i + \mu_t + \varepsilon_{i,t} \tag{2}$$

(2) Spatial Error Model (SEM): the spatial spillover effect formed by the region is caused by random impact. The change of a factor not only has a certain impact on the research object itself (direct effect), but also on its surrounding regions (indirect effect).

Control variables are used to represent the impact of adjacent areas on the local human settlements and to investigate whether the impact of spatial spillover is positive or negative.

$$Y_{i,t} = \alpha + \beta X_{i,t} + C_i + \mu_t + \nu_{i,t} \tag{3}$$

$$\nu_{i,t} = \lambda \sum_{j=1}^{N} W_{i,j} \nu_{i,t} + \varepsilon_{i,t} \tag{4}$$

(3) Spatial Durbin Model (SDM): it is also called panel spatial interaction model. Endogenous and exogenous interactions among regions and the error terms with autocorrelation jointly form the spatial dependence of geographical elements.

$$Y_{i,t} = \alpha + \rho \sum_{j=1}^{N} W_{i,j} Y_{i,t} + \beta X_{i,t} + \theta \sum_{j=1}^{N} W_{i,j} X_{i,t} + C_i + \mu_t + \varepsilon_{i,t} \tag{5}$$

In (2)–(5), $Y_{i,t}$ is the explained variable, $i$ is the number of spatial regions, and $t$ is the time dimension ($i$ = 1, 2, ... , $N$; $t$ = 1, 2, ... , $T$). $X_{i,t}$ is the exogenous explanatory variable matrix of $n*k$. $\beta$ is the regression coefficient of the explanatory variable in the form of $k*1$ dimensional coefficient vector. $\rho$ is spatial autocorrelation coefficient with a value between −1 and 1, which is used to describe the interaction between explained variable $Y_{i,t}$ and that of adjacent units. $W_{i,j}$ is the non-negative space weight matrix of $n*n$. $\varepsilon_{i,t}$ is the error term, whose value is (0, $\sigma^2$) and $\lambda$ is the spatial autocorrelation coefficient of random error term.

Lagrange Multiplier Error, Lagrange Multiplier Lag, Robust-Lmlag and Robust-Lmerror are used to judge the specific form of the model. If LM cannot reject SLM and SEM, Wald tests (including Wald-Lag and Wald-Error) need to be further used to determine whether SLM or SEM should be adopted. If both tests reject the original hypothesis, we should use SDM. Finally, the Hausman test is used to judge whether to use a fixed effect model or random effect model. Meanwhile, Likelihood Ratio (LR) is used to judge whether to use individual fixed effect and time fixed effect.

## 3. Results

### 3.1. Spatial-Temporal Distribution Characteristics of Human Settlements

3.1.1. Weight Calculation

Air quality index (AQI) is not limited to natural systems. According to index attributes, the original data of different properties are dimensionless, and the weight values of various factors of human settlements are calculated (Table 4). The score of them is obtained according to the corresponding weight and the grade is divided according to the value.

3.1.2. Spatial State Mode

ArcGIS is used to combine the research and calculation data with geographical space and analyze the spatial state mode and pattern evolution of human settlements according to seven administrative geographical divisions. It is also used to study spatial differentiation patterns in Chinese cities every year. In terms of basic spatial pattern, North China is always in the inferior area of human settlements, while Southern China is a prefecture-level city, which is in the advantageous area of high-level human settlements all year round. On the whole, the average level order of urban human settlements in the seven regions of China is Southern China (0.7310) > Southwest China (0.6608) > East China (0.6515) > Northeast China (0.6496) > Northwest China (0.6049) > Central China (0.5901) > North China (0.5565) (Figure 2). The distribution characteristics of human settlements are not consistent with those of air quality index.

**Table 4.** Weight of comprehensive indicators.

| Index Level | 2013 Comprehensive Weight | 2014 Comprehensive Weight | 2015 Comprehensive Weight | 2016 Comprehensive Weight | 2017 Comprehensive Weight | 2018 Comprehensive Weight |
|---|---|---|---|---|---|---|
| Annual average temperature (°C) | 0.019186 | 0.015905 | 0.015588 | 0.016971 | 0.015306 | 0.017485 |
| Average annual relative humidity (%) | 0.028799 | 0.024098 | 0.023433 | 0.025708 | 0.024022 | 0.026126 |
| Average annual rainfall (mm) | 0.061549 | 0.061281 | 0.057241 | 0.065213 | 0.054655 | 0.022764 |
| Mean wind speed (m/s) | 0.048002 | 0.049271 | 0.049399 | 0.049417 | 0.046642 | 0.049372 |
| AQI | 0.042333 | 0.055785 | 0.066665 | 0.075073 | 0.069676 | 0.062174 |
| $PM_{2.5}$ ($\mu g/m^3$) | 0.054970 | 0.042038 | 0.047260 | 0.051424 | 0.050467 | 0.048745 |
| $PM_{10}$ ($\mu g/m^3$) | 0.046133 | 0.038082 | 0.048433 | 0.053159 | 0.047379 | 0.050701 |
| $NO_2$ ($\mu g/m^3$) | 0.016445 | 0.012690 | 0.017277 | 0.016062 | 0.012890 | 0.015858 |
| $SO_2$ ($\mu g/m^3$) | 0.017858 | 0.015636 | 0.017651 | 0.019439 | 0.018522 | 0.017570 |
| $CO$ ($\mu g/m^3$) | 0.015268 | 0.016656 | 0.013529 | 0.016141 | 0.015839 | 0.014444 |
| $O_3$ ($\mu g/m^3$) | 0.029677 | 0.025952 | 0.033510 | 0.034874 | 0.033105 | 0.019983 |
| Green and cover rate in the built-up area (%) | 0.038226 | 0.030201 | 0.040336 | 0.037270 | 0.028553 | 0.024926 |
| Per capita park green area ($m^2$) | 0.026577 | 0.026536 | 0.024790 | 0.024517 | 0.028778 | 0.032346 |
| Number of days with good or above Grade 2 air quality (days) | 0.125913 | 0.157345 | 0.178139 | 0.126340 | 0.175582 | 0.203186 |
| Industrial smoke (powder) dust treatment rate (%) | 0.119979 | 0.118696 | 0.108977 | 0.110473 | 0.114188 | 0.118164 |
| Highway passenger Volume (10,000 persons) | 0.014979 | 0.016717 | 0.012565 | 0.013426 | 0.015028 | 0.013418 |
| Highway freight volume (10,000 tons) | 0.010766 | 0.016967 | 0.009485 | 0.010096 | 0.016781 | 0.015835 |
| Number of operating vehicles (vehicles) with bus (electric) vehicles | 0.011740 | 0.011242 | 0.013353 | 0.012783 | 0.011584 | 0.012609 |
| Construction of urban housing (10,000 square meters) | 0.022987 | 0.021975 | 0.015328 | 0.015610 | 0.024970 | 0.016155 |
| The second output value accounted for the GDP proportion (%) | 0.076449 | 0.083266 | 0.074329 | 0.078510 | 0.054512 | 0.075008 |
| Urban population density (people/$km^2$) | 0.102396 | 0.092155 | 0.076169 | 0.075524 | 0.086972 | 0.077324 |
| Total gas supply (artificial and natural gas) (ten thousand cubic meters) | 0.006516 | 0.006760 | 0.006070 | 0.006361 | 0.006031 | 0.006240 |
| Total LPG gas supply (ton) | 0.008423 | 0.008166 | 0.009590 | 0.009891 | 0.008388 | 0.011817 |
| Dust industrial dust emission per capita (ton) | 0.017259 | 0.021747 | 0.021214 | 0.024329 | 0.018918 | 0.020891 |
| Per capita industrial sulfur dioxide emissions (ton) | 0.037569 | 0.030831 | 0.019666 | 0.031389 | 0.021214 | 0.026860 |

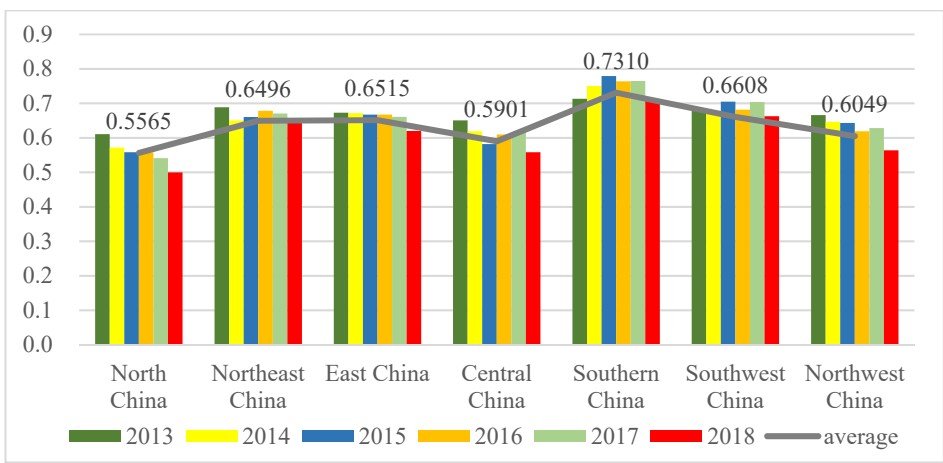

**Figure 2.** The average level of urban human settlements in the seven regions of China.

ArcGIS natural discontinuity method is used to classify data and form five types of human settlements in prefecture-level cities in China: low level, lower level, medium level, high level, and higher level (Figure 3).

In the past six years, the Hebei, Shanxi, Shandong, and Henan provinces have accounted for the largest number of cities with low-level urban human settlements. There are no low-level cities in the coastal provinces south of Shandong. Although the overall score decreased in 2018, the low-level cities classified in this year accounted for 8.8% of the cities in China, the least in six years, followed by 9.15% in 2017 and up to 10.92% in 2014. In the research time series, the lower-level cities are mainly concentrated in Shaanxi, Gansu, Ningxia, and the Central Plains, whose distribution was the least in 2016 and the most in 2015 and 2017. A total of 60 cities account for 21.13% of China. The urban medium-level has almost been the largest number of distribution in the past six years, but the overall trend is declining, gradually decreasing from 31.69% to 27.82%. Most of them are located in Northern China and the provinces flowing through the trunk of the Yangtze River to the east of the Sichuan Province. High-level cities are classified, and concentrated provinces begin to move southward, with the majority of provinces and regions south of the Yangtze River. In terms of time change, 2015 accounted for 21.13%, which is the lowest year, increasing to both sides. The most occurred in 2018, with 86 cities accounting for 30.28%. The proportion of high-level urban human settlements is relatively small, only above the low-level, which increased from 13.38% in 2013 to 17.96% in 2015. It decreased to 13.73% with the lowest value in six years in 2017. It is mainly distributed in coastal provinces to the south of Zhejiang and Heilongjiang Provinces.

### 3.1.3. Pattern Evolution

(1) Global spatial autocorrelation

The *Moran's I* estimate of the comprehensive level of urban human settlements in 2013–2018 are all positive (Figure 4). The test results in 6 years show that *p*-value is about 0.0010 and *z*-value > 1.96, showing significant results. The level of urban human settlements presents spatial positive correlation. Areas with the comprehensive level of human settlements as "HH (first quadrant)" or "LL (third quadrant)" are concentrated and distributed in space, meaning the high value is surrounded by the high value and low value cities are also low value scoring cities.

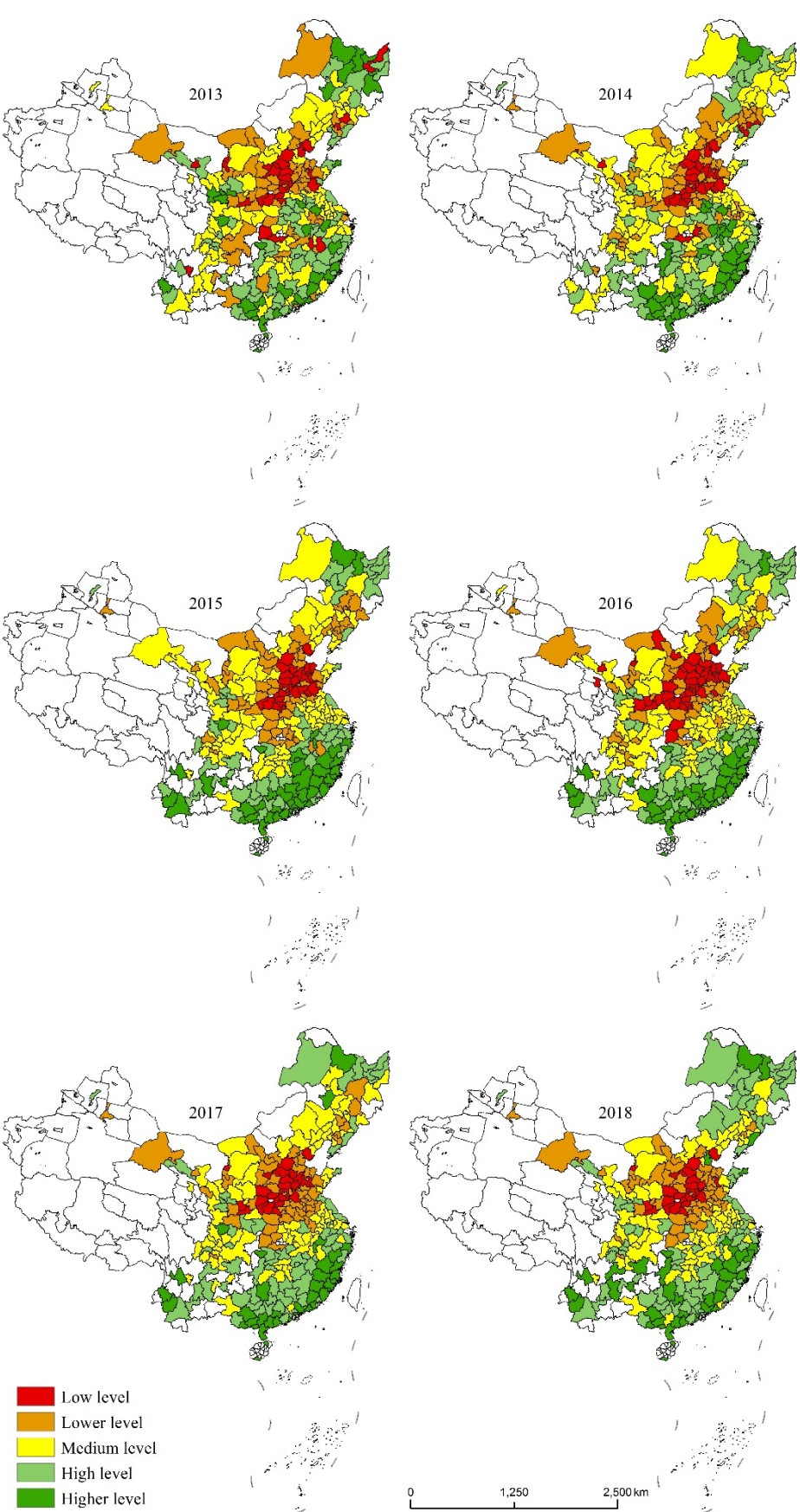

**Figure 3.** Spatial-temporal evolution for comprehensive scores of human settlements in China's prefecture-level cities.

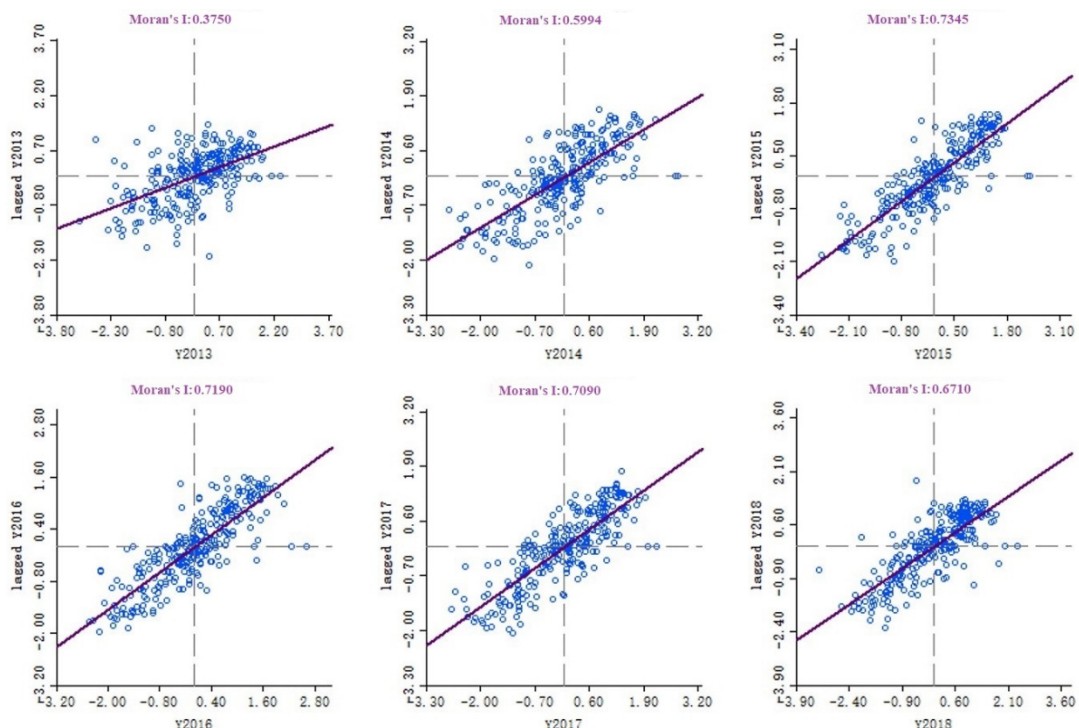

**Figure 4.** Scatter plot of *Moran's I* index in China's prefecture-level human settlements from 2013 to 2018.

From 2013 to 2015, the *Moran's I* value generally increased (Table 5), the spatial distribution of human settlements showed a stronger HH and LL agglomeration pattern, and it also showed a trend of increasing spatial agglomeration. The spatial differences of urban human settlements were also gradually obvious. From 2015 to 2018, the spatiality was reduced, and the level of human settlements in urban spatial units showed a certain convergence, which is becoming increasingly prominent. It can be seen that after 2015, the human settlements based on air quality in Chinese cities no longer exist in a single individual form. The correlation between cities is becoming closer and closer.

**Table 5.** *Moran's I* estimate of the level of human settlements from 2013 to 2018.

| Year | *Moran's I* | E(I) | *p*-Value | *z*-Value |
|------|-----------|------|-----------|-----------|
| 2013 | 0.3750 | −0.0035 | 0.0010 | 9.6651 |
| 2014 | 0.5994 | −0.0035 | 0.0010 | 15.0378 |
| 2015 | 0.7345 | −0.0035 | 0.0010 | 19.8175 |
| 2016 | 0.7190 | −0.0035 | 0.0010 | 18.2767 |
| 2017 | 0.7090 | −0.0041 | 0.0010 | 18.5267 |
| 2018 | 0.6710 | −0.0044 | 0.0010 | 16.2060 |

(2) Local spatial pattern

From 2013 to 2018, the High-High cluster of urban human settlements was mainly distributed in the southern region, and concentrated in the Heilongjiang province in some years, which was relatively continuous in space. Low-Low cluster areas were mainly distributed in the Shanxi, Hebei, Shandong, and Henan provinces, and the cluster distribution area was large. The number of Low-High clusters was small, with 1–2 cities each year, mainly distributed in the Anhui and Jiangsu provinces. Except for 2016, there were no High-Low cities. There were five cities in other years, which were distributed in the Gansu, Hebei, Jiangsu, and Henan provinces. It can be seen that the spatial distribution of human settlements based on air quality in Chinese cities is more dependent, and the number of cities significantly dependent is gradually increasing. The number of Not Significant cities gradually decreased from 199 to 152 (Figure 5).

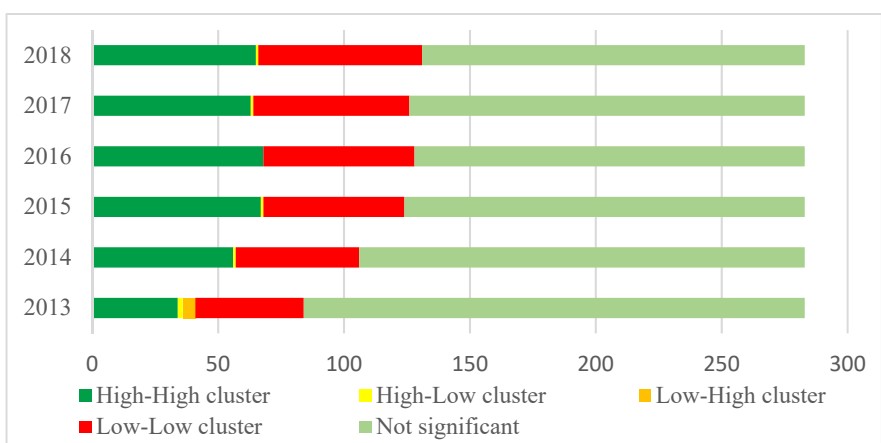

**Figure 5.** Statistics on the number of *Local Moran's I* of cities from 2013 to 2018.

According to the spatial-temporal evolution of urban human settlements from 2013 to 2018 (Figure 6), the spatial pattern remains unchanged as a whole, but some areas change between different years. Local spatial autocorrelation shows that the "cold spots" with low levels of human settlements are continuously distributed in the North China Plain in time and space, while the "hot spots" cities are continuously distributed in Southern China. Due to the natural factors, a large number of High-High agglomeration cities have been distributed in the Guangdong, Guangxi, Zhejiang, Fujian, and Yunnan provinces, and scattered into a whole. Due to the low urban economy and population density and less energy consumption, Heilongjiang has a low score in human settlements where human factors affect urban air quality. Because of the guidance of policy and the improvement of environmental protection awareness, the agglomeration of High-High cities is becoming increasingly obvious and the number of cities is also increasing. At the same time, Low-Low cities are also gradually increasing, spreading from Beijing, Tianjin, Hebei, and Shanxi to most cities in Henan, Shandong, Shaanxi, and Ningxia. North China and the Central Plains are densely populated with large energy consumption, rapid economic development, and urban construction, so human factors have a great impact on air quality. Secondly, the vegetation coverage of the natural environment is relatively low, and the number of days of air pollution is more than that of southern cities. Therefore, there is a significant agglomeration in this area. From 2013 to 2018, the number of cities in the two categories of spatial evolution pattern increased, showing a convergent distribution. The number of Not Significant cities gradually decreased.

### 3.2. Factors Affecting the Distribution Difference of Human Settlements

The mechanism factor of urban human settlements by traditional measurement methods assumes that the research units are independent of each other. Due to the existence of spatial autocorrelation, the spatial measurement model is used to calculate the spatial effect of human settlements, and the comprehensive score is taken as the explained variable (Y). The explanatory variables are: scientific and technological investment (X1), per capita GDP (X2), urbanization rate (X3), education level (X4), advanced industrial structure (X5), urban average elevation (X6), and resident activity (X7) [51–55]. T = 6, N = 283. Matlab 2012 is used for estimation. The testing process and results of three types of models with fixed effect and random effect are as follows.

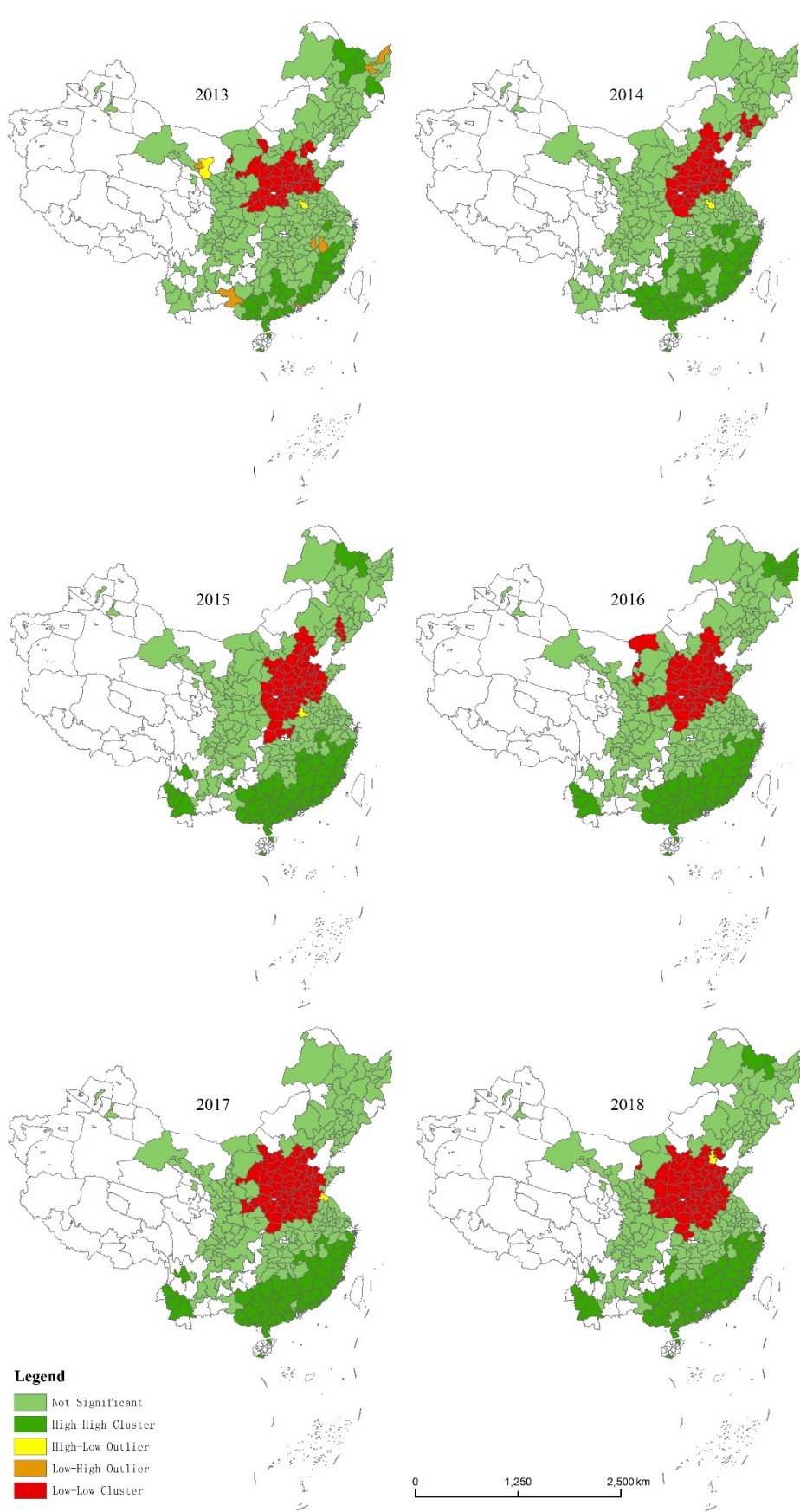

**Figure 6.** *Anselin Local Moran's I* map of human settlements in China from 2013 to 2018.

In the fixed effect spatial lag panel data model (Table 6), the impact of five explanatory variables (X1-science and technology investment, X2-per capita GDP, X3-urbanization rate, X5-advanced industrial structure, X6-urban average elevation) on urban human settlements is significant in the statistical data. In the random effect spatial lag panel data model, the impact of four explanatory variables (X1-science and technology investment, X2-per capita GDP, X3-urbanization rate, X4-education level) is significant. The spatial lag model of fixed effect and random effect have passed the maximum likelihood LM Lag test and LM Error test, indicating that there is a spatial correlation impact on urban human settlements. Both fixed effect and random effect spatial lag models have passed the Robust LM Error test. In the fixed effect linear regression model, *p*-values of science and technology investment, per capita GDP, urbanization rate, urban average elevation and residents' activity are greater than 0.05. It indicates that the three regression coefficients in this model are not significant.

**Table 6.** Estimation results of spatial lag evaluation model for urban human settlements characteristics.

| Variables | Fixed Effect | | | Random Effect | | |
|---|---|---|---|---|---|---|
| | Coefficient | *t*-Stat | Probability | Coefficient | *t*-Stat | Probability |
| lnX1 | 0.001165 | 1.223659 | 0.221081 | 3145.404 | −0.243600 | 0.807541 |
| lnX2 | 0 | −0.143361 | 0.886005 | −0.000215 | 0.197614 | 0.843347 |
| lnX3 | −0.000059 | −0.994290 | 0.320082 | 0 | 1.180926 | 0.237632 |
| lnX4 | −0.000031 | −5.252817 | 0 | 0.000046 | −1.138767 | 0.2548 |
| lnX5 | −0.015320 | −7.157567 | 0 | −0.000009 | −1.037222 | 0.299633 |
| lnX6 | 0 | 0.01399 | 0.988838 | −0.002590 | 3.320827 | 0.000898 |
| lnX7 | 0.001305 | 1.725711 | 0.084399 | 0.000013 | 1.781365 | 0.074853 |
| W*dep.var. | 0.998985 | 3945.387 | 0 | 0.002114 | 304.5005 | 0 |
| teta | - | - | - | 0.265423 | 17.24327 | 0 |
| $R^2$ | | 0.6728 | | | 0.8886 | |
| Sigma$^2$ | | 0.0025 | | | 0.0009 | |
| log-likelihood | | 2585.6269 | | | 3145.4038 | |
| LMlag | | 1781.3148 | | | 40,034.6297 | |
| R-LMlag | | 8029.8153 | | | 71,8143.7069 | |
| LMerror | | 71.323 | | | 4.0723 | |
| R-LMerror | | 6319.8235 | | | 678,113.1495 | |

On the whole, LM test values of the fixed effect and random effect models under specific matrix are positive, and most of them pass the 10% significance level test, indicating that the results are obvious. Therefore, the existence of residual spatial autocorrelation has been confirmed. Spatial autoregressive coefficient ρ (the coefficient of W*dep.var) and the estimated value of variable X1 of urban human settlements are positive, and both have passed the 1% significance probability test. It fully shows that there is a positive spatial correlation between China's human settlements in each city. The coefficient of W*dep.var with fixed effect shows that the spillover effect is obvious. From the adjusted $R^2$, Sigma$^2$, log-likelihood, the fixed effect spatial lag panel model is significantly weaker than that of the random effect. Meanwhile, the spatial autoregressive coefficient of the fixed effect lag model is significantly greater than the sub regression coefficient of the random effect lag model.

The spatial lag model has good fitting in both fixed effect and random effect, and the fitting of random effect is better than that of fixed effect. It is in line with the actual considerations. In the data analysis of urban human settlements, it should be considered that human settlements of Chinese cities are greatly affected by surrounding cities. So, the impact of space on the city cannot be ignored when analyzing urban human settlements.

By analyzing the regression results of the model, it can be seen that the spatial error models of fixed effect and random effect pass the maximum likelihood LM Lag test and LM Error test, indicating that there is an obvious spatial correlation impact on urban human settlements (Table 7). Moreover, the spatial error models of the two effects pass the Robust LM Error test, indicating that there is obvious autocorrelation in the spatial error term. The

significance of regression parameters can also provide relevant reference for the selection of models to a certain extent. In the fixed effect spatial error model, the *p*-values of per capita GDP and urbanization rate are greater than 0.05, indicating that the two regression coefficients are not significant. In the spatial error model of random effect, the *p*-values of science and technology investment, per capita GDP, urbanization rate, education level, and advanced industrial structure are greater than 0.05, indicating that the two regression coefficients are not significant. From the statistical data of the model test, the spatial error model has better interpretation in the evaluation of urban human settlements.

**Table 7.** Estimation results of spatial error evaluation model for urban human settlements characteristics.

| Variables | Fixed Effect | | | Random Effect | | |
|---|---|---|---|---|---|---|
| | Coefficient | *t*-Stat | Probability | Coefficient | *t*-Stat | Probability |
| lnX1 | −0.002480 | −2.334380 | 0.019576 | −0.001050 | −1.078920 | 0.280624 |
| lnX2 | 0 | −0.375060 | 0.707617 | 0 | −0.190200 | 0.849149 |
| lnX3 | −0.000016 | −0.269210 | 0.787766 | 0.000049 | 1.173063 | 0.24077 |
| lnX4 | −0.000020 | −3.519320 | 0.000433 | −0.000003 | −0.312250 | 0.75485 |
| lnX5 | −0.017970 | −8.161420 | 0 | −0.002380 | −0.417340 | 0.676432 |
| lnX6 | 0.000011 | 2.862237 | 0.004207 | 0.00029 | 16.3351 | 0 |
| lnX7 | −0.002130 | −2.390450 | 0.016828 | 0.00605 | 2.044096 | 0.040944 |
| spat.aut. | 0.988972 | 334.0992 | 0 | 0.996368 | 23658.92 | 0 |
| teta | - | - | - | 88.19199 | 13.41322 | 0 |
| R$^2$ | | −0.011800 | | | 0.8881 | |
| Sigma$^2$ | | 0.0025 | | | 0.0009 | |
| log-likelihood | | 2604.1892 | | | 2620.315 | |
| LMlag | | 3702.4477 | | | 1191.641 | |
| R-LMlag | | 347.4313 | | | 1250.0513 | |
| LMerror | | 4933.6623 | | | 0.0002 | |
| R-LMerror | | 1578.6458 | | | 58.4105 | |

The LM test values under the spatial error model of fixed effect and random effect are positive, and most of them pass the 10% significance test, indicating that the results are significant. Autocorrelation coefficient λ is positive and the estimated value of variable X1 is negative, which all pass the 1% significance probability test. The coefficient of W*dep.var of fixed effect shows that the spillover effect is obvious, or the spillover of this city to other cities is obvious. From the adjusted R$^2$, Sigma$^2$, log-likelihood, the fixed effect spatial error panel model is significantly better than the random effect spatial lag panel model. The spatial autoregressive coefficient of the fixed effect lag model is significantly less than the sub regression coefficient of the random effect lag model.

The fixed effect of the spatial error model shows that urban human settlements tend to affect the city's scientific and technological investment, economic development, urbanization, and urban natural advantages. However, for the upgrading of industrial structure, it has little influence. The results of random effect show that the driving mechanism has a great influence on the city's scientific and technological investment, economic development, urbanization, and education. For the urban natural advantages of the city, the influence is small.

In order to comprehensively and accurately analyze the spatial effects of urban human settlements, the fixed effect and random effect are analyzed by using the Spatial Durbin Model (SDM). It includes spatial weights of explanatory variables and explained variables (Table 8). When testing its spatial effect, the spatial auto-regressive coefficient W*dep.var. of SDM is significantly positive when the significance level is 10% (0.993996). Most of the spatial lag coefficients of dependent variables are negative and most of them fail to pass the significance test at the 1% level. Under the random effect, the spatial auto-regressive coefficient W*dep.var. of SDM is significantly positive (0.967977) when the significance level is 10%. Most of the spatial lag coefficients of the dependent variables are negative and most of them fail to pass the significance test of the 1% level. This indicates that

there is no obvious spatial correlation on the human settlements in various regions. That is, the level of human settlements in a region does not depend on the level of that in adjacent regions and its explanatory variables. For both fixed effect and random effect, only when the significance level test with the advanced degree of industrial structure exceeds 1% and the coefficient is negative, can they have a significant inhibitory effect on urban human settlements.

**Table 8.** Estimation results of Durbin evaluation model for urban human settlements characteristics.

| Variables | Fixed Effect | | | Random Effect | | |
|---|---|---|---|---|---|---|
| | Coefficient | *t*-Stat | Probability | Coefficient | *t*-Stat | Probability |
| lnX1 | −0.002466 | −2.361442 | 0.018204 | −0.001563 | −1.698671 | 0.089381 |
| lnX2 | 0.000000 | 0.420666 | 0.673999 | 0.000000 | 0.016323 | 0.986977 |
| lnX3 | −0.000029 | −0.496513 | 0.619532 | −0.000008 | −0.181877 | 0.855680 |
| lnX4 | −0.000024 | −4.116232 | 0.000039 | −0.000013 | −1.636228 | 0.101792 |
| lnX5 | −0.020358 | −9.184800 | 0.000000 | −0.017493 | −4.818972 | 0.000001 |
| lnX6 | 0.000028 | 5.333164 | 0.000000 | 0.000017 | 1.615204 | 0.106266 |
| lnX7 | −0.001899 | −2.066503 | 0.038781 | −0.002697 | −1.738743 | 0.082080 |
| W*lnX1 | 0.013040 | 4.546095 | 0.000005 | 0.002786 | 0.983582 | 0.325321 |
| W*lnX2 | 0.000000 | 2.654114 | 0.007952 | 0.000000 | 2.240181 | 0.025079 |
| W*lnX3 | −0.000462 | −2.641054 | 0.008265 | −0.000162 | −1.505127 | 0.132291 |
| W*lnX4 | −0.000078 | −2.628386 | 0.008579 | 0.000055 | 1.822676 | 0.068352 |
| W*lnX5 | −0.016386 | −2.115604 | 0.034379 | 0.027305 | 4.767539 | 0.000002 |
| W*lnX6 | −0.000046 | −4.802428 | 0.000002 | −0.000016 | −0.927091 | 0.353879 |
| W*lnX7 | 0.013126 | 5.716435 | 0.000000 | 0.008057 | 2.475201 | 0.013316 |
| W*dep.var. | 0.993996 | 657.950793 | 0.000000 | 0.967977 | 129.876552 | 0.000000 |
| teta | | | | 0.285798 | 17.304632 | 0.000000 |
| R$^2$ | | 0.699600 | | | 0.888400 | |
| Sigma$^2$ | | 0.002300 | | | 0.000900 | |
| log-likelihood | | 2650.950500 | | | 3164.152300 | |
| Wald_spatial_lag | | 157.679600 | | | 57.688700 | |
| Wald_spatial_error | | 107.286800 | | | 31.766700 | |
| LR_spatial_ lag | | 130.124000 | | | 888.309000 | |
| LR_spatial_error | | 83.633500 | | | 1938.500000 | |
| Hanuman | | | | | 58.286900 | |

The four factor coefficients of the fixed effect of SDM are negative, showing a negative correlation. The impact of per capita GDP is small, and residents' activity shows a positive correlation with human settlements. Among the seven explanatory variables under random effect, only the advanced industrial structure exceeds the 1% significance-level test and its coefficient is negative. That is, this factor has a significant inhibitory effect on urban human settlements. While the other six variables exceed the 1% significance-level test, of which only urban elevation shows a positive correlation with human settlements.

From the decomposition of the above results (Table 9), in the fixed effect model, the coefficients of direct effect, indirect effect, and overall effect of per capita GDP and education level fail to pass the 1% significance test. This shows no significant spatial spillover. In the random effect model, the per capita regional GDP, urbanization rate, education level, and average elevation fail to pass the 1% significance test. They have little relationship with the development of adjacent cities and do not play an obvious role in human settlements.

**Table 9.** The evaluation results of the effects of Durbin model.

| Effect Evaluation | Variables | Fixed Effect | | | Random Effect | | |
|---|---|---|---|---|---|---|---|
| | | Coefficient | *t*-Stat | *t*-Prob | Coefficient | *t*-Stat | *t*-Prob |
| Direct effects | lnX1 | 0.000388 | 0.306805 | 0.759217 | −0.001340 | −1.173813 | 0.241453 |
| | lnX2 | 0.000000 | 2.085903 | 0.037880 | 0.000000 | 1.194471 | 0.233290 |
| | lnX3 | −0.000163 | −2.135358 | 0.033587 | −0.000040 | −0.796255 | 0.426549 |
| | lnX4 | −0.000051 | −4.325309 | 0.000021 | −0.000005 | −0.481062 | 0.630843 |
| | lnX5 | −0.030116 | −9.598687 | 0.000000 | −0.015463 | −4.234122 | 0.000031 |
| | lnX6 | 0.000023 | 4.938440 | 0.000001 | 0.000017 | 1.842039 | 0.066512 |
| | lnX7 | 0.001129 | 1.074293 | 0.283603 | −0.001550 | −1.000729 | 0.317810 |
| Indirect effects | lnX1 | 0.800067 | 3.862730 | 0.000139 | 0.041324 | 0.450830 | 0.652456 |
| | lnX2 | 0.000007 | 2.622878 | 0.009190 | 0.000001 | 1.924636 | 0.055274 |
| | lnX3 | −0.037285 | −2.762278 | 0.006114 | −0.005359 | −1.290572 | 0.197902 |
| | lnX4 | −0.007724 | −3.092455 | 0.002183 | 0.001243 | 1.326532 | 0.185729 |
| | lnX5 | −2.779643 | −4.522011 | 0.000009 | 0.311432 | 2.093364 | 0.037204 |
| | lnX6 | −0.001384 | −3.365275 | 0.000870 | −0.000007 | −0.024911 | 0.980143 |
| | lnX7 | 0.861756 | 5.040209 | 0.000001 | 0.170666 | 1.906720 | 0.057566 |
| Total effects | lnX1 | 0.800455 | 3.851552 | 0.000145 | 0.039984 | 0.433206 | 0.665194 |
| | lnX2 | 0.000007 | 2.622373 | 0.009203 | 0.000001 | 1.922912 | 0.055491 |
| | lnX3 | −0.037448 | −2.764135 | 0.006081 | −0.005399 | −1.290961 | 0.197767 |
| | lnX4 | −0.007774 | −3.099903 | 0.002130 | 0.001237 | 1.310436 | 0.191107 |
| | lnX5 | −2.809759 | −4.554184 | 0.000008 | 0.295968 | 1.977671 | 0.048932 |
| | lnX6 | −0.001361 | −3.325550 | 0.000998 | 0.000009 | 0.030815 | 0.975439 |
| | lnX7 | 0.862885 | 5.032566 | 0.000001 | 0.169116 | 1.882057 | 0.060851 |

## 4. Discussion

### 4.1. Analysis of Spatial-Temporal Variation and Influence Factors

There are various research indicators of human settlements at home and abroad, and the evaluation is based on a certain system for practical problems [56,57]. With practical problems and people-oriented theory, this paper selects air as evaluation perspective and reorganizes the research index system of human settlements. However, in recent years, most studies are carried out at urban agglomeration, provinces, and cities [58]. The large-scale human settlement units are related to the built-up area environment, with few studies focusing on index evaluation [59,60]. Therefore, combined with the air perspective based on circulation and strong linkage, this paper makes an innovative and targeted study on human settlements of prefecture-level cities in China.

(1) Horizontal comparison of time shows that there is little difference in the average comprehensive score of human settlements in 6 years. The gap between the minimum and maximum scores in 2013 is the smallest. By 2018, the overall human settlements of all cities had declined. Looking at them in prefecture-level cities in China, the human settlements in North China, Henan, Hubei, and Shandong provinces have always been below the medium level. Human settlements in most parts of Southwest, East and South China and Northeast Heilongjiang have always been maintained at a high level. The spatial differentiation state studied in this paper presents a spatial pattern of low in the middle and high around. The change difference of urban level in the North–South is greater than that in the East–West, which is roughly equivalent to the overall conclusion of Li's research on human settlements [61].

Low level cities are concentrated in Shaanxi, Gansu and Ningxia and the Central Plains, and less in the Northwest; although there is sand and dust weather and less green coverage in Northwest China, the human factors, such as building dust and traffic emission caused by low urban development and low population density, have little impact on the comprehensive score of human settlements. It is different from the basic pattern conclusion of Zhang's comprehensive index evaluation of cities, which increases step by step from west to east [62]. High-level cities are classified intensively, and most of them are provinces

and regions south of the Yangtze River. Although they account for a small proportion of all prefecture-level cities, the state of spatial pattern is gradually improved from north to south. It is basically consistent with the research conclusion of Li [63].

(2) Compared with the conclusion of spatial state model and relevant air quality studies, the spatial regional distribution of high concentration values of different pollutants studied by sun is different [64] but similar to Tang's conclusion on China's spatial-temporal differentiation [27]. They all have strong North–South differences.

Firstly, this is not only because of air pollutants but also because the central and eastern regions have a large population base, rapid economic and social development, more and more energy consumption, higher pollutant values, and residents' intuitive feelings about traffic, building dust, and industrial pollution emissions. They all make the overall human settlements of cities in the central region the lowest. Secondly, the population is large in most cities of the eastern region and the per capita park green space area is smaller than that of northern cities. The treatment rate of industrial smoke (powder) dust is low, which is mostly the same as Wang's contribution to the air [65].

(3) The human settlements explored from the perspective of air also show the same positive autocorrelation as $PM_{2.5}$, and the local spatial clustering is significant [52]. With a scatter chart, the global *Moran's I* index under this study shows a spatial positive correlation in the past six years, and the degree of correlation is relatively strong. At the comprehensive development level, it has the spatial convergence of high-value and low-value clusters. The number of Low-Low agglomeration area and High-High area are increasing, but the proportion in the first quadrant (H-H) is the largest. It indicates that the convergence effect of urban development in geographical space is obvious.

(4) In terms of pollutant distribution, economic and social development, and natural conditions, human settlements between different cities in China gradually show an obvious polarization. However, most cities show insignificant correlation with adjacent regions. Local spatial autocorrelation shows that Southern China is the "hot spot" of human settlements, and the North China Plain is the "cold spot" area. They are consistent with the significant high-value agglomeration results of Yang's multi-perspective comprehensive study [66] and the hot spot agglomeration areas of high air pollution studied by Xiao [67]. However, the middle and lower reaches of the Yellow River, the Yangtze River Delta, Pearl River Delta, and other places involved in scholars' studies do not show significant agglomeration in this paper. In addition, the intermittent small-scale high-level agglomeration areas in Heilongjiang are rarely seen in previous studies.

(5) Air pollution assessment and socio-economic development mostly select air measurement models to analyze the influencing factors [68]. This paper draws on relevant research of scholars and selects the influencing factors of human settlements. According to the relevant data of the model test, there is a significant spatial positive correlation between science and technology investment in SDM with fixed effect and random effect. The per capita regional GDP has little impact on human settlements. The urbanization rate, the degree of advanced industrial structure, and the urban average elevation have a certain spatial spillover and show a negative correlation. Among the seven explanatory variables, science and technology investment and the degree of advanced industrial structure have the greatest impact. It can be seen that urban upgrading is very important for human settlements.

(6) The construction time of human settlements in many cities at home and abroad shows that social and economic development play a basic role in promoting the construction and development of urban human settlements. If human settlements are strong, the economic strength is also good. However, the human settlements score based on air quality does not completely depend on the economic conditions. Therefore, in order to effectively improve urban human settlements, it is necessary to improve the government-led "top-down" and public-led "bottom-up" regulation system. We should formulate laws and policies suitable for social and economic development to fully tap into the potential of each city in order to enhance the strength of the city.

### *4.2. Limitations*

Due to less available data, time series research is short. In addition, this paper is an innovative attempt to select and integrate indicators of human settlements and influencing factors. Therefore, there are still some deficiencies in index system and scale, and there is a lack of empirical research.

In addition, when there are more systematic indicators, the research methods, such as spatial autocorrelation and spatial measurement, indicator variables, and measurement models used at different scales are different. Subjective evaluation and practical application should be considered and more comprehensive methods should be used for evaluation and test in combination with multi-source data, such as big data.

### 5. Conclusions

The level of China's human settlements determined by air-orientation presents a spatial pattern of low in the middle and high around, and has strong spatial correlation. Under the condition of balanced development of various systems, we should focus on the human settlements in the middle and lower reaches of the Yellow River. We should also try to build a hierarchical and focused evaluation model, as well as a characteristic urban planning and linkage development pattern. We should improve the urban development model under the concept of creating a clean air city, increase investment in science and technology, optimize the industrial structure, and improve social environmental equity and residents' well-being.

However, residents with different subjective attributes have different subjective understandings of the environment and corresponding behavior. For a certain population, there is a certain regularity statistically. Therefore, in the process of future evaluation, we should pay attention to public participation and reflect the public's psychological feelings on environmental quality. We should quantify the qualitative indicators through residents' scores, improve human settlement evaluation systems, and their impact mechanism at various time and space scales. With empirical experiments in different regions with a long time series, we can provide a theoretical basis for regional human settlements under different development conditions.

**Author Contributions:** Conceptualization, X.L.; methodology, S.L. and H.L.; software, S.L. and H.L.; validation, X.L.; formal analysis, S.T.; investigation, Y.G.; resources, Y.G.; data curation, S.L. and H.L.; writing—original draft preparation, S.T. and S.L.; writing—review and editing, Y.G. and H.L.; visualization, S.L. and Y.G.; supervision, all Authors; project administration, X.L.; funding acquisition, X.L. and S.T. All authors have read and agreed to the published version of the manuscript.

**Funding:** This research was funded by the National Natural Science Foundation of China (grant number 41671158); Liaoning Province Natural Science Foundation Project (grant number 2020-BS-182).

**Data Availability Statement:** The data presented in this study are available on request from the corresponding author.

**Conflicts of Interest:** The authors declare no conflict of interest.

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
