# Peer review of "Air Quality and the Spatial-Temporal Differentiation of Mechanisms Underlying Chinese Urban Human Settlements"

_land, doi:10.3390/land10111207_

Round 1

Reviewer 1 Report

The paper is dedicated to formulation the innovative evaluation model of human settlements. The paper is informative and good structured; the introduction and literature review provide sufficient background.

Author Response

Thank you for your letter and for the reviewers’ comments concerning our manuscript entitled “Research on the Spatial and Temporal Differentiation and Mechanism of Chinese Urban Human Settlements Guided by Air Quality” (land-1435748). 

Special thanks to you for your good comments. We tried our best to improve the manuscript and made some changes in the manuscript. These changes will not influence the content and framework of the paper.Revised portion are marked in red and highlight in the paper.

Reviewer 2 Report

The authors present a methodology to investigate the spatiotemporal distribution of human settlements based on air conditions. It is an interesting approach, but several amendments are required to be presented better. Therefore, I suggest accepting the manuscript after major revisions.  

  • General: there are many long sentences that are hard to follow.
  • Table 2: Please explain the symbols of the column Criterion attribute
  • Line 120: introduce CRITIC and AHP.
  • Lines 120-122: the sentence is unclear
  • Table 3:
  • have you calculated these weights? If so, why do you present this table in this section instead of Results section?
  • What is each level index (1st column)?
  • Line 142: what do you mean the panel data model?
  • Figure 5: explain NO abbreviation and generally all the abbreviations in the text.
  • Line 274: explain “T=7, N=283”
  • Line 275: “In the table, there are statistics such as R2, Sigma2, CS, log likelihood” à explain, what’s is their use, what their values denote.
  • Table 7: please define the variables X9-X14
  • Line 415: first time cold and hot spots are mentioned. They should have been mentioned earlier.

Author Response

Thank you for your letter and for the reviewers’ comments concerning our manuscript entitled “Research on the Spatial and Temporal Differentiation and Mechanism of Chinese Urban Human Settlements Guided by Air Quality” (land-1435748). Those comments are all valuable and very helpful for revising and improving our paper, as well as the important guiding significance to our researches. We have studied comments carefully and have made correction which we hope meet with approval. Revised portion are marked in red in the paper. The main corrections in the paper and the responds to the reviewer’s comments are as flowing:

  • Point 1: General: there are many long sentences that are hard to follow.

Response 1: We have made correction according to the Reviewer’s comments.(in highlight)

  • Point 2: Table 2: Please explain the symbols of the column Criterion attribute

Response 2: The nature of each indicator is relative to the evaluation target, + refers "positive", and higher values mean the better. - refers "negative", and lower values mean the better. * refers moderation, and moderate values are fine. When the index value is less than the moderate value, it conforms to positive index. Moreover, when it is greater than the moderate value, it conforms to inverse index.(line 176-180,in red)

  • Point 3: Line 120: introduce CRITIC and AHP.

Response 3: Considering the Reviewer’s suggestion, we have introduced CRITIC and AHP in line 146-164(in red).

  • Point 4: Lines 120-122: the sentence is unclear

Response 4: We have re-written this part according to the Reviewer’s suggestion.(line 166-173, in red)

  • Point 5: Table 3:have you calculated these weights? If so, why do you present this table in this section instead of Results section?

Response 5: Considering the Reviewer’s suggestion, we have presented this table in  Results section,(line 232-240, in red)

  • Point 6: What is each level index (1st column)?

Response 6: We have made correction according to the Reviewer’s comments(line 239,in red).

  • Point 7: Line 142: what do you mean the panel data model?

Response 7: We have re-written this according to the Reviewer’s suggestion.(line198)

  • Point 8: Figure 5: explain NO abbreviation and generally all the abbreviations in the text.

Response 8: As Reviewer suggested that, we re-written all abbreviation in the text and figure.(in red)

  • Point 9: Line 274: explain “T=7, N=283”

Response 9: We are very sorry for our incorrect writing, and we have made correction.(line 353). T means year, N means number of spatial regions(we explained it in line 215-222)

  • Point 10: Line 275: “In the table, there are statistics such as R2, Sigma2, CS, log likelihood” à explain, what’s is their use, what their values denote.

Response 10: Considering the Reviewer’s suggestion, we have explained these in line 223-229, deleted CS, and line 371-382, line 406-414 were added in part 3.2.(in red)

  • Point 11: Table 7: please define the variables X9-X14

Response 11: Line 422, the statements of “ X9-X14” were corrected as “W*lnX1-W*lnX7”.

  • Point 12: Line 415: first time cold and hot spots are mentioned. They should have been mentioned earlier.

Response 12: Considering the Reviewer’s suggestion, we have mentioned them in line 323-325.

Special thanks to you for your good comments. We tried our best to improve the manuscript and made some changes in the manuscript. These changes will not influence the content and framework of the paper.

Reviewer 3 Report

This is a very difficult paper to read and comprehend. I see potentiality in the study and the idea is interesting but the presentation of data and methodologies is flawed. I don't think it is only a language problem (some checks, however, are necessary to correct longer sentences). I believe it is a problem of rationale and organization of the paper. I would suggest you to rethink the structure and the consequentiality of the paper focusing on:

i) a broader and comprehensive discussion;

ii) a broader literature review.

iii) true aims and scope of this study.

iv) future research directions.

v) generalization of this study to broader contexts

vi) pros & cons of the methodology.

Thank you.

Author Response

Thank you for your letter and for the reviewers’ comments concerning our manuscript entitled “Research on the Spatial and Temporal Differentiation and Mechanism of Chinese Urban Human Settlements Guided by Air Quality” (land-1435748). Those comments are all valuable and very helpful for revising and improving our paper, as well as the important guiding significance to our researches. We have studied comments carefully and have made correction which we hope meet with approval. Revised portion are marked in red in the paper. The main corrections in the paper and the responds to the reviewer’s comments are as flowing:

  • Point 1:a broader and comprehensive discussion;

Response 1:We have re-written this part according to the Reviewer’s suggestion. And line 468-474, line 490-492, line 530-540 were added.(in red)

  • Point 2:a broader literature review.

Response 2:Line 46-72, were added.(in red)

  • Point 3:true aims and scope of this study.

Response 3:We have re-written this part according to the Reviewer’s suggestion.(line 116-127)

  • Point 4:future research directions.

Response 4:We have explained future research directions in line 561-569.(in red)

  • Point 5:generalization of this study to broader contexts

Response 5:Considering the Reviewer’s suggestion, we have re-written this part according to the Reviewer’s suggestion.(line 32-45, in red)

  • Point 6:pros & cons of the methodology.

Response 6:Considering the Reviewer’s suggestion, we have introduced

(1)pros of CRITIC and AHP in line 146-164, spatial econometric model in line 195-197.(in red)

(2)cons of the methodology were added in line 544-550.(in red)

Special thanks to you for your good comments. We tried our best to improve the manuscript and made some changes in the manuscript. These changes will not influence the content and framework of the paper.

Round 2

Reviewer 2 Report

Paper's quality has been significantly improved. Therefore, I suggest accepting in the current form.

Reviewer 3 Report

Good revision overall. I have a last comment, and sorry if I insist on this point. I don't like the title, I suggest a shorter formulation:

Air quality and the spato-temporal Differentiation of Mechanisms underlying Chinese Urban Human Settlements

Thank you.